# High-speed three-dimensional photoacoustic computed tomography for preclinical research and clinical translation

Li Lin [1,4], Peng Hu [1,4], Xin Tong[1,4], Shuai Na[1,4], Rui Cao[1], Xiaoyun Yuan [1,2], David C. Garrett[1], Junhui Shi [1,3], Konstantin Maslov[1] & Lihong V. Wang [1✉]

Photoacoustic computed tomography (PACT) has generated increasing interest for uses in preclinical research and clinical translation. However, the imaging depth, speed, and quality of existing PACT systems have previously limited the potential applications of this technology. To overcome these issues, we developed a three-dimensional photoacoustic computed tomography (3D-PACT) system that features large imaging depth, scalable field of view with isotropic spatial resolution, high imaging speed, and superior image quality. 3D-PACT allows for multipurpose imaging to reveal detailed angiographic information in biological tissues ranging from the rodent brain to the human breast. In the rat brain, we visualize whole brain vasculatures and hemodynamics. In the human breast, an in vivo imaging depth of 4 cm is achieved by scanning the breast within a single breath hold of 10 s. Here, we introduce the 3D-PACT system to provide a unique tool for preclinical research and an appealing prototype for clinical translation.

[1] Caltech Optical Imaging Laboratory, Andrew and Peggy Cherng Department of Medical Engineering, California Institute of Technology, Pasadena, CA, USA. [2] Present address: Department of Electronic Engineering, Tsinghua University, Haidian District, Beijing, China. [3] Present address: Zhejiang Lab, China Artificial Intelligence Town, Hangzhou Zhejiang, China. [4] These authors contributed equally: Li Lin, Peng Hu, Xin Tong, Shuai Na. ✉email: LVW@caltech.edu

Photoacoustic (PA) computed tomography (PACT) is a noninvasive hybrid imaging modality that combines the functional optical contrast of diffuse optical tomography with the high spatial resolution of ultrasonography[1]. When a short-pulsed laser irradiates biological tissues, the light energy is partially absorbed by the tissue and converted into heat. Subsequently, a pressure rise induced by transient thermoelastic expansion propagates as a wideband ultrasonic wave (referred to as PA wave) through the biological tissue. The PA waves are then detected by ultrasonic transducers placed around the tissue surface to reconstruct the optical absorption distribution in the tissue[2]. The sensitivity to optical absorption in biological tissues provides PACT-rich contrast mechanisms relating to various intrinsic and extrinsic chromophores, enabling structural, functional, and molecular imaging[3]. In addition to the rich optical contrast, the conversion from light to ultrasound also provides this imaging technique with ultrasonically defined spatial resolution at depths beyond the ~1-mm optical diffusion regime, which limits the penetration capability of ballistic optical imaging modalities[4].

This hybrid nature grants PACT unique advantages and makes it complementary to other mainstream imaging modalities in clinical practice: (1) compared with pure optical imaging techniques such as fluorescence imaging in humans[5], PACT can sustain high spatial resolution in the deep tissue. (2) compared with ultrasonic imaging, PACT has rich intrinsic and extrinsic optical contrasts and is free of speckle artefacts[4]; (3) compared with X-ray computed tomography (X-ray CT) and positron emission tomography, PACT is free of radioactive material and uses nonionizing illumination; (4) compared with magnetic resonance imaging (MRI), laser-based PACT provides higher spatial resolution at a lower cost, though it faces a challenge to penetrate the entire human body. Specifically, the cutting-edge PACT system we are presenting in this manuscript can image the human breast in 10 s with full penetration (4 cm) and nearly isotropic spatial resolution (0.37–0.39 mm) that is finer than breast 3T-MRI (0.5 mm in-plane, 1.3 mm thick)[6] or a typical 7T-functional MRI (fMRI) (isotropic 0.75 mm)[7].

Employing different light illumination and detection schemes, several PACT systems with three-dimensional (3D) field-of-view (FOV) have been developed for preclinical studies[8–11] and clinical practice[12–18]. Although these systems have advanced PACT performance, key limitations remain unaddressed. Specifically, the current systems' limitations mainly arise from their shallow imaging depth, slow imaging speed, and/or compromised image quality owing to the limited noise-equivalent sensitivity[19] and FOV. Here, we define high image quality as sufficient contrast-to-noise ratio (CNR) to reveal detailed structures on the order of the inherent spatial resolution (i.e., high clarity) within an FOV large and deep enough to cover the target region. A useful rule of thumb is that a CNR of 2.2 can be detected with a confidence level of 98%[20].

To provide a large view aperture (i.e., view angle)[21], some researchers utilized spherical detection matrices, which, however, were designed and implemented differently[8–10,12–16]. For example, Deán-Ben et al.[9] recently reported a PACT system with 256 transducer elements integrated on a partial cap for small animal imaging. This design can generate a small volumetric image with a single laser shot, but its well-resolved FOV is theoretically limited to a few millimeters (e.g., diameter <4 mm) according to the spatial Nyquist sampling criterion[22]. Subsequently, the authors scanned the system helically around a mouse body to densely sample the surface tissue and generated an elegant superficial image. However, without scanning along the radial direction, the mouse's internal organs were not imaged clearly. The limited-view aperture (i.e., <2π steradian solid angle) further compromised the image quality when a higher imaging speed was required for brain imaging[8].

For human breast imaging, Matsumoto et al.[12] developed a PACT system using a sparse hemispherical detector array that scanned in a spiral pattern on a plane. The dense sampling produced high-quality vascular images near the skin surface. Similar to the aforementioned system, however, the well-resolved FOV of the detector array could only recover a small breast volume and the system could not image the deep tissue well without an elevational scan. In addition, the illumination wavelengths they used suffer from higher scattering in the breast and a lower safety limit for maximum permissible exposure (1/5 of the limit for 1064-nm light)[23]. Accordingly, the imaging depth of the system was limited to the superficial region of the breast. More recent work published by Oraevsky et al.[13] and Schoustra et al.[14] employ scanned arc ultrasonic arrays for human breast imaging. Both studies showed neither cross-sectional breast images at different depths nor 3D-rendered views, presumably owing to the limited imaging depth. Similarly, the sparse spatial sampling limited the well-resolved FOV in breasts, thus compromising the image quality while also requiring a long scanning time of several minutes. In addition, the scanning of the laser beam varied the light energy distribution,[14] which violates the assumption of 3D image reconstruction algorithms[24,25] that light energy distribution remains consistent during scanning. Other than the systems that utilized spherical detection matrices, researchers also explored 3D imaging by detecting with a planar sensor[17] or scanning a linear array[18]. Though the systems were simpler, the spatial resolution was anisotropic and the image quality was reduced owing to the limited-view aperture.

In this paper, we developed a PACT system, which produces volumetric images of biological tissues with superior imaging depth, image quality, and speed. Rather than being dedicatedly used for a single application, we have demonstrated its in vivo imaging versatility in the rat brain and human breast, representing two extreme imaging scales. In the rat brain, we clearly imaged angiographic anatomies and functions from the cortex to the Circle of Willis. In the human breast, the system can generate a volumetric image with a penetration up to 4 cm by scanning the breast within a single breath-hold of 10 s. Our in vivo imaging results distinguish 3D-PACT as a cutting-edge PA imaging system that meets all the following conditions: (1) deep penetration and scalable FOV to accommodate imaging objects from the rat brain to human breast, (2) isotropically high spatial resolution within the FOV in 3D space, (3) high noise-equivalent sensitivity to reveal small structures deep in tissues, (4) high imaging speed to minimize motion artefacts and enable functional studies, and (5) minimal limited-view artefacts. Accordingly, 3D-PACT could provide high-quality images with functional optical contrast and high imaging speed to directly benefit both preclinical research and clinical practice, which have been highlighted by increasing interest from clinical researchers and practitioners.

## Results

**Characteristics of the 3D-PACT.** The 3D-PACT system is shown in Fig. 1a. An imaging platform is placed above the ultrasonic array housing to support the imaging subject. A disposable holding cup (plastic wrap) is clamped on the imaging aperture to separate the imaging system from the subject. The 3D-PACT system is composed of five modules: (1) four arc-shaped ultrasonic transducer arrays are integrated in a hemispherical housing (Fig. 1b, c). Each array has 256 transducer elements with a central frequency of 2.25 MHz and a one-way bandwidth of more than 98% (Supplementary Fig. 1). (2) 1024-channel pre-amplification circuits are attached to the base of the ultrasonic array housing (Fig. 1b, Supplementary Fig. 2). Four phase plates transmit amplified PA signals to four data acquisition (DAQ) modules. (3)

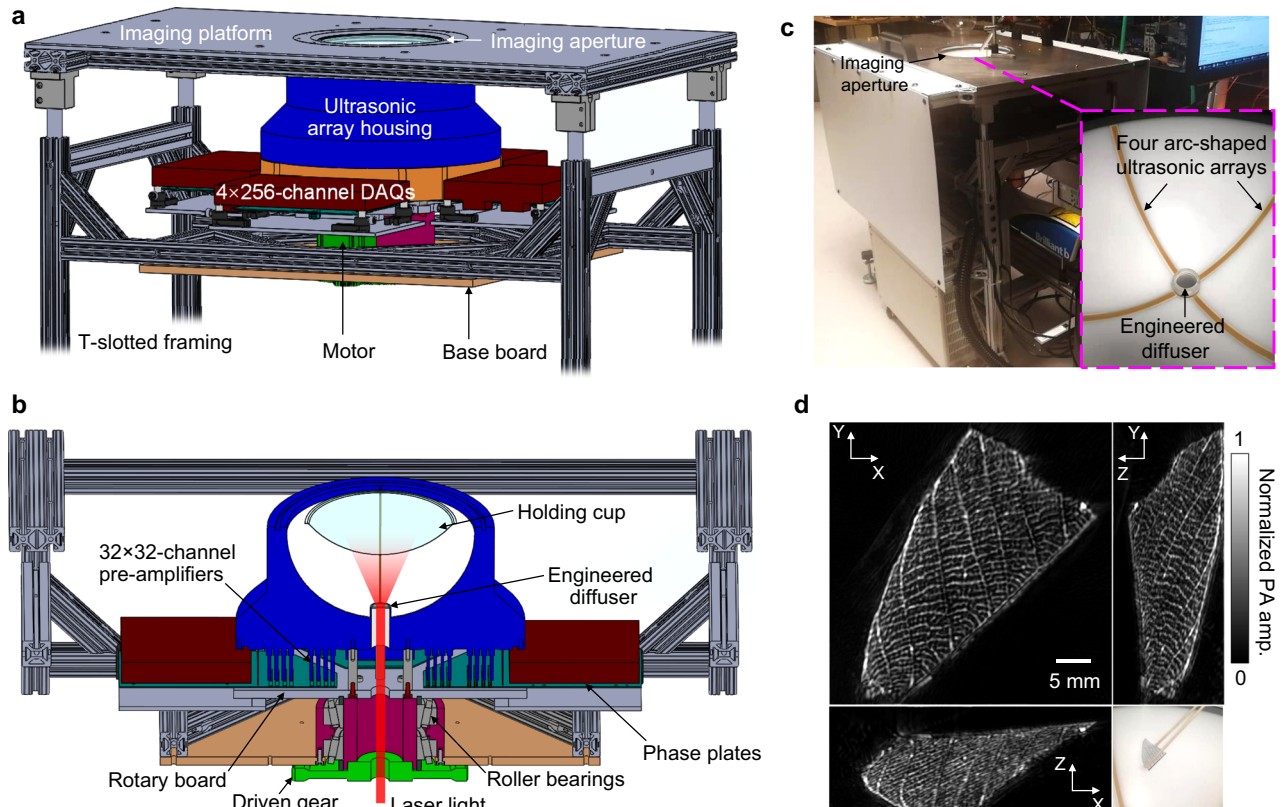

**Fig. 1 Representations and photographs of the 3D-PACT system and the leaf skeleton images. a** Perspective view of the system. DAQ, data acquisition module. **b** Cut-away view of the system with the imaging platform removed. **c** Photograph of the system with a close-up view of the ultrasonic arrays. **d** Photograph and maximum amplitude projection (MAP) images of a leaf skeleton reconstructed after geometric calibration (see Methods).

Each DAQ module has 256-channel analog-to-digital converters with a tunable gain from 12 to 51 dB. (4) A mechanical scanning module rotates the aforementioned three modules coaxially for 90 degrees to form a densely-sampled hemispherical detection matrix. (5) The light beam is directed to the subject from an Nd: YAG laser through a one-inch lens tube glued on the bottom of the hemispherical housing (Fig. 1b, c). Two engineered diffusers are sealed in the lens tube to expand the beam for illumination with redundancy for safety (Methods and Supplementary Fig. 3); the laser beam would be expanded even if one of the diffusers was broken or misaligned during imaging. We take advantage of the low optical attenuation of 1064 nm light to achieve deep optical penetration in biological tissues[26].

Owing to manufacturing defects in the transducer arrays, which are slightly imperfect circular arcs, the reconstructed images of a leaf skeleton phantom are slightly blurred (Supplementary Fig. 4). To correct these defects, we measured a point source placed at multiple positions and applied Jacobi iteration to find the true geometric position of each transducer element (Methods and Supplementary Fig. 4). An improvement in image quality was achieved and the detailed leaf skeletons were clearly revealed (Fig. 1d). After calibrating the imaging system, we further imaged a 50-µm-diameter aluminum particle, allowing us to experimentally quantify the spatial resolution in 3D space (390 µm on $X$–$Y$ plane, 370 µm along $Z$ direction) (Supplementary Fig. 5). The scanning time to generate a volumetric image ranges from 2 to 10 s depending on the targeted FOV[22].

**3D-PACT of the rat brain anatomy and functions**. Small animals are principal models for preclinical studies, and the imaging

of their brains has an important role in neuroscience research. 3D-PACT is capable of providing high-quality images of the whole rat brain anatomy within 5 s or can continuously monitor the cerebral functions with a volumetric imaging rate of 0.5 Hz, which can be further increased by utilizing a laser with higher repetition rate (e.g., 5 Hz for a 100-Hz laser).

As shown in Supplementary Fig. 6, the rat head was mounted horizontally with the cortical region attached to the center of the membrane. Although the top part of the skull was thinned (see Methods) to acquire detailed angiograms of the brain by reducing the attenuation and distortion of the skull on the PA waves, the skull remained intact for functional imaging. The 2.25-MHz center frequency allows 3D-PACT to produce high-quality images with tolerable acoustic distortion from the rat skull[27]. Taking advantage of the deep penetration of 1064 nm light and the panoramic acoustic detection, a volumetric image of the rat whole brain (10 mm in depth) was acquired with detailed vasculature (Fig. 2a and Supplementary Movie 1). The image was acquired by averaging eight 5-second scans and subsequently reconstructed using a dual-speed-of sound 3D back-projection algorithm (see Methods). Fig. 2b, c show cross-sectional images of the brain on sagittal and coronal planes, respectively. Supplementary Fig. 7 shows the brain images with comparable clarity acquired in 5 s without averaging.

Serving as the center of the nervous system, the brain coordinates neural activities which, as in fMRI, can be related to cerebral hemodynamics[28,29]. Imaging at a 0.5-Hz volumetric frame rate, we used 3D-PACT to monitor the hemodynamics while modulating inhaled oxygen (Fig. 3a–c) and anesthesia content (Fig. 3d–f). Specifically, we measured the hemodynamics in four major vessels (Fig. 3c, f, and Supplementary Fig. 8) during the two processes to demonstrate the advantages of 3D-PACT's spatiotemporal

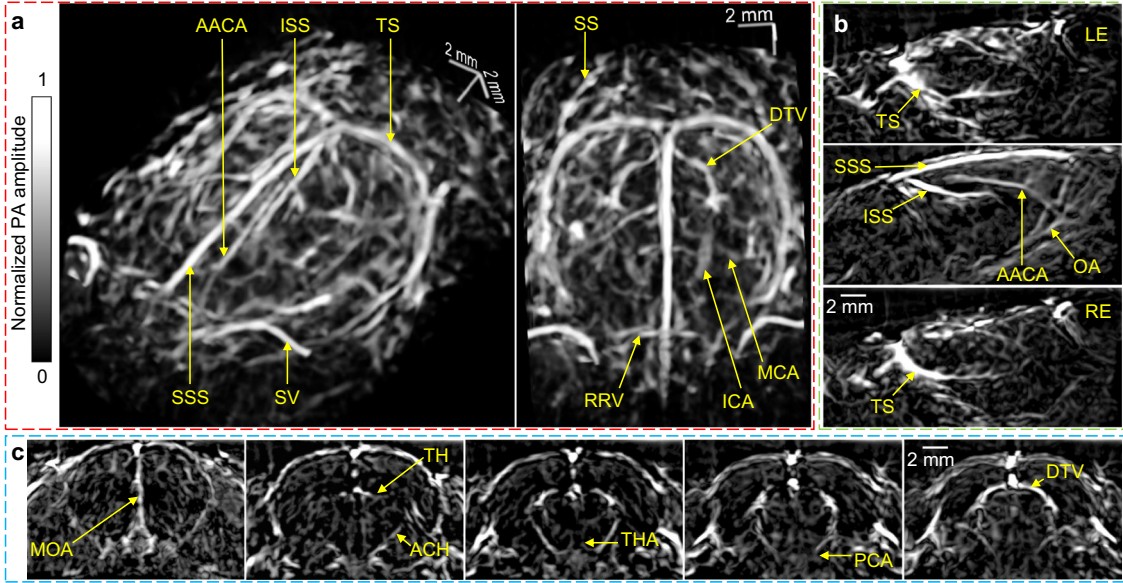

**Fig. 2 3D-PACT of a rat brain in vivo. a** Perspective angiograms of the rat whole brain (Supplementary Movie 1). **b** Cross-sectional images of the brain on different transverse planes. **c** Cross-sectional images of the brain on different coronal planes. *AACA* azygos of the anterior cerebral artery, *ACH* anterior choroidal artery/vein, *DTV* dorsal thalamic vein, *ICA* internal carotid artery, *ISS* inferior sagittal sinus, *LE* left eye, *MCA* middle cerebral artery, *MOA* medial orbitofrontal artery, *OA* olfactory artery, *PCA* posterior cerebral artery, *RE* right eye, *RRV* rostral rhinal vein, *SS* sigmoid sinus, *SSS* superior sagittal sinus, *SV* supraorbital vein, *TH* transverse hippocampal artery/vein; *THA* thalamoperforating artery, *TS* transverse sinus.

resolution. As oxyhemoglobin dominates the optical absorption at 1064 nm (i.e., $\mu_{a(oxyhemoglobin)} \approx 10 \times \mu_{a(deoxyhemoglobin)}$, $\mu_a$: absorption coefficient)[26], the changes in spatially averaged PA signals mainly reflect the variation of oxyhemoglobin content within the region of interest (ROI) over time. For instance, to measure hemodynamics in major blood vessels, the ROIs were selected slightly larger than the vessels' cross-sections, thus both the oxygen saturation (sO2) and the blood volume (or vascular diameter) would affect PA signal amplitudes. Furthermore, we also measured the intrinsic functional connectivity (Fig. 3g) and the hemodynamics induced by the electrical stimulations to the front limbs (Fig. 3h).

To observe the brain's hemodynamic response to hypoxic challenges, we manipulated the oxygen concentration in the inhalation gas (see Methods) to during imaging. In Fig. 3a, we selected three cross-sections on the coronal plane from bregma to lambda and observed a global decrease in the PA signal in response to 30 s of nitrogen inhalation. Fig. 3b shows the spatially averaged PA amplitude changes in the circled regions in Fig. 3a.

During hypoxic challenges, we noticed that the PA signal variations in the four vessels were comparable (Fig. 3c and Supplementary Fig. 8a)[30]. We also observed larger fractional changes in PA signals from the four major vessels compared with brain tissue (Fig. 3b), which includes both blood vessels and brain cells. To explain this difference, we surmise that in Fig. 3b, the time-invariant PA signals from lipid, water, and proteins[31] moderated the change in PA signals from hemoglobin. Another observation is that the slopes of the PA signals plateaued during the 40-second hypoxia in Fig. 3b, c, which is likely owing to a reduced oxygen consumption rate in response to extended hypoxia. In comparison, the slopes of the PA signals are more consistent during the 10-second hypoxia. This agrees with the expectation that oxygen consumption rate is more stable at the beginning of hypoxia[32].

In addition to hypoxic challenges, we measured PA signal changes from deep to light anesthesia (Fig. 3d–f). Isoflurane, a well-known vasodilator[33], can also reduce the cerebral oxygen extraction fraction[34,35]. Therefore, the decrease in PA signals within different brain regions from deep to light anesthesia

(Fig. 3d, e) could be a combined effect of reduced blood volume and sO2.

Moreover, we also monitored hemodynamics in the same major vessels during the awakening process (Fig. 3f and Supplementary Fig. 8b). Similar hemodynamics were observed in the two arteries and two veins. To interpret this similarity, we surmise that in veins, both the decrease in sO2 and the contraction of vascular diameter[35] contribute to the PA signals' decrease. In arteries, although sO2 is expected to be more stable during the awakening process[35], the vascular contraction may be more pronounced than that in veins[35]. Another observation is that the decrease in PA signals from major vessels (Fig. 3f) was comparable to that in brain tissues (Fig. 3e). To explain this, our hypothesis is that in Fig. 3e, although the time-invariant PA signals from lipid, water, and proteins could moderate the changes in PA signals from hemoglobin, small vessels in the brain tissue are expected to contract more than major vessels during the awakening process[35].

Like fMRI[36], we used 3D-PACT to assess the intrinsic functional connectivity across spatially separated brain regions through spontaneous fluctuations in PA signals (particularly during resting state) by measuring the spontaneous hemodynamic responses in cerebral regions (Supplementary Movie 2). We identified 20 functional regions across the whole brain and computed the correlation coefficients of every pair (see Methods). Fig. 3g shows clear correlation between the contralateral regions in left and right brain hemispheres. The high volumetric imaging rate also allows us to measure faster cerebral hemodynamics (Supplementary Movie 2). In Fig. 3h, we observed higher PA amplitude changes in the corresponding motor sensory regions when the rat's front limbs were separately stimulated by periodic electrical pulses (see Methods). We further plotted the PA amplitude changes in the motor sensory regions in response to the electrical stimulation.

Our anatomical and functional brain imaging results demonstrate the potential of 3D-PACT as an imaging tool with deep penetration, isotropic resolution, and high speed for studying rat

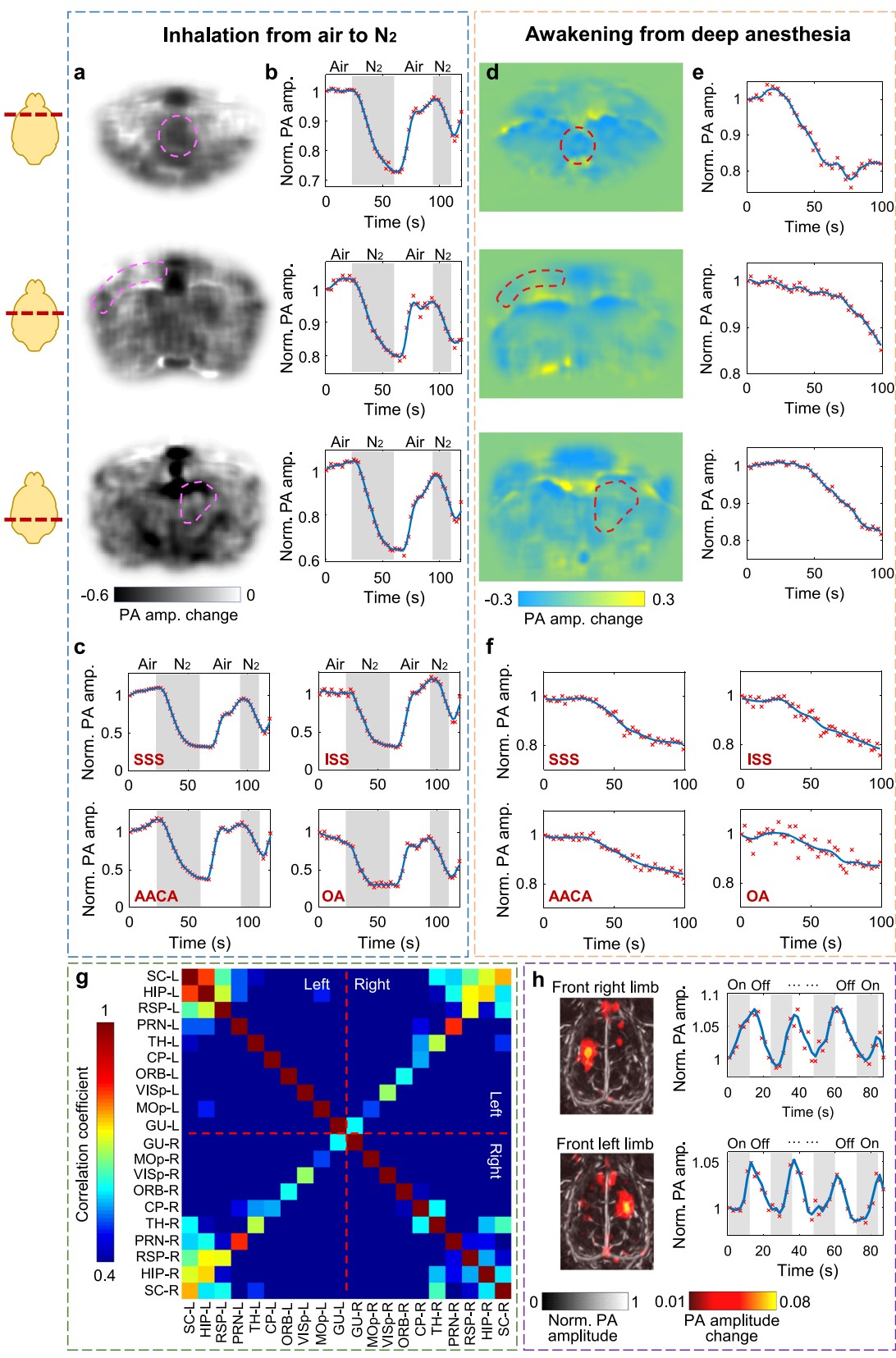

whole brain structures and functions, which were previously challenging to accomplish using optical contrast. For small animal functional imaging, whereas the imaging speed of 3D-PACT (0.5-Hz volumetric imaging rate) is not yet substantially faster than fMRI (9–11 slices in 2 s)[37], the high spatial resolution of 3D-PACT enabled measurement of hemodynamics

in individual vessels. Although the rat brain can be fixed in place during imaging using a nose cone and tooth bar, artefacts induced by periodic motions (e.g., breathing) can be largely removed by time-gated motion correction (see Methods). We have demonstrated this technique by imaging the abdomen of a pregnant rat, whose embryo was clearly revealed after correcting the respiratory

**Fig. 3 Functional rat brain imaging. a** PA amplitude changes in different brain cross-sections after the rat inhaling pure nitrogen ($N_2$) for 30 s. **b** PA amplitude changes in the circled regions in **a**. **c** PA amplitude changes in four major arteries and veins during hypoxic challenges. *AACA* azygos of the anterior cerebral artery, *OA* olfactory artery, *SSS* superior sagittal sinus, *ISS* inferior sagittal sinus. **d** PA amplitude changes in different brain cross-sections while awakening from deep anesthesia. **e** PA amplitude changes in the circled regions in **d**. **f** PA amplitude changes in the same four major vessels while awakening from deep anesthesia. **g** Correlation matrix of 20 functional regions across the whole brain. Notice the correlation between left and right hemispheres. **h** Brain cortical hemodynamics in response to the electrical stimulation to the front limbs. *CP* caudoputamen, *GU* gustatory area, *HIP* hippocampal region, *MOp* primary motor area, *PRN* pontine reticular nucleus, *ORB* orbital area, *RSP* retrosplenial area, *SC* superior colliculus, *TH* thalamus, *VISp* primary visual area.

motion (Supplementary Fig. 9). Time gating can also be used to reconstruct fast but periodic movements (e.g., heart beating). Accordingly, in addition to brain imaging, 3D-PACT can be used to reveal structural, functional, and molecular information in many other parts of small animals.

**3D-PACT of human breasts in vivo.** Multiple large prospective clinical trials have demonstrated the importance of early detection in improving breast cancer survival[38,39]. In addition, more recent studies have shown that tumor biology also has an important role in the prognosis of an individual cancer[40,41]. An imaging modality with high sensitivity and specificity which is cost effective and safe to use would provide an important unmet medical need[42,43].

For breast imaging, PACT can offer high spatial and temporal resolutions with sufficiently deep nonionizing optical penetration[44]. In biological tissue, the optical absorption contrast of PACT is much higher than mammographic X-ray contrast[45]. However, the 1/e attenuation coefficient for 1064-nm light in an average breast is ~0.9 cm$^{-1}$ [26,46], which is slightly higher than that for X-rays (0.5–0.8 cm$^{-1}$)[47]. In the near-infrared (NIR) region, hemoglobin is the principal optical absorber to provide PACT an endogenous contrast (i.e., blood vessel density and morphology, total hemoglobin concentration, oxygenation state, etc.), which is closely correlated with the tumor growth and metastasis[48–50]. The optical absorption contrast can also serve as an indicator of the breast cancer's response to treatment[51,52].

To demonstrate the 3D-PACT of breast imaging, we imaged the two breasts of a 31-year-old healthy female volunteer. Her breast size was 36 C, which is the average size in the United States[53]. The 3D-PACT system was placed underneath a dedicated imaging bed[44] with minimal separation. With the human subject lying prone on the bed, the breast to be imaged was slightly compressed against the chest wall by the holding cup to reduce the optical tissue thickness from the nipple to the chest wall (Supplementary Fig. 6). By scanning the transducer arrays around the breast within a single breath hold of 10 s, we clearly revealed the blood vessels from the nipple to the chest wall (Fig. 4, Supplementary Movie 3 and 4). The cross-sectional images in Fig. 4b, d show detailed angiographic structures at different depths up to 4 cm, visualizing the vasculature down to an apparent vascular diameter of 400 μm. In our breast images, most large blood vessels are distributed near the skin surface (i.e., <2 cm deep, mainly derive from superior, lateral, and internal thoracic vessels), whereas blood vessels deeper in the breast (i.e., intercostal perforators) tend to be smaller and perpendicular to the chest wall. Such a distribution complies with the general angiographic anatomy of the human breast[54,55]. To further test the sensitivity of detecting small abnormalities in the breast, we imaged a breast-mimicking phantom with tumors of different sizes embedded at 2 cm depth (Supplementary Fig. 10). We made the tumor and breast phantoms with similar optical properties as in real human breasts[56]. The absorption coefficient of the tumors ($\mu_a = 0.105$ cm$^{-1}$) was 2.1 times that of the breast phantom ($\mu_a = 0.05$ cm$^{-1}$)[56,57]. The reduced scattering coefficient $\mu'_s$ of the

phantoms was 5 cm$^{-1}$ at 1064 nm[26,57]. The system is sufficiently sensitive to detect a 1 mm-diameter tumor embedded in the breast-mimicking phantom.

**Discussion**

We have developed a 3D-PACT system that reveals detailed angiographic structures and functions with deep penetration, isotropic spatial resolution in scalable FOV, and high imaging speed. The massively parallel acoustic detection and one-to-one mapped low-noise amplifiers and DAQ circuits enable 3D imaging of an entire rat brain in 2 s and a human breast in 10 s. The high imaging speed allows 3D functional imaging of the whole rat brain, including the analysis of intrinsic functional connectivity and the measurement of hemodynamics while modulating inhaled oxygen, anesthesia content, and electrical stimulations. The large view aperture (i.e., ≈2π steradian solid angle) provides nearly isotropic spatial resolution and high image quality. In addition, by combining the 1064-nm light illumination with 2.25-MHz ultrasonic detection, 3D-PACT achieved up to 4 cm in vivo imaging depth in human breast, which is expected to be sufficient to image a D cup breast with a painless compression against the chest wall (over 99% of the US population have breast sizes of D cup or smaller[53]). Furthermore, 3D-PACT requires neither ionizing radiation nor an exogenous contrast agent, yielding zero risk to the imaging subject. The high imaging speed makes the 3D-PACT even more appealing.

The modular design also makes the 3D-PACT system easy to be maintained and upgraded. With the current hardware configuration, 3D-PACT has limitations in its single-wavelength illumination and mechanical scanning time (2–10 s). These limitations could be mitigated by replacing certain modules in the system: (1) using multi-wavelength illumination to provide additional imaging contrast[58] (e.g., indocyanine green); (2) using lasers with higher repetition rates to image a larger FOV during the same scanning time or image the same FOV within shorter time; (3) replacing the stepper motor with a servomotor with higher torque to allow faster mechanical scanning; (4) using lasers with higher pulse energy to create a larger illumination area for breast imaging. For example, to improve the speed for small animal functional imaging, one could use a faster laser (e.g., LPY742-100, 100 Hz repetition rate, Litron Lasers Ltd.) and a servomotor with higher torque (e.g., EMG-50ASA, 3384 oz-in torque, Anaheim Automation Inc.). The system could then scan at 5 Hz for functional 3D imaging. One could also couple an OPO laser (e.g., SpitLight 1000 OPO, InnoLas Laser, Inc.) for multi-contrast imaging.

It is likely feasible to image awake rodents using 3D-PACT, but the system may need to employ ultrasonic arrays with smaller footprints (i.e., diameter) for easy operation. Inspired by PA microscopy of the awake mouse brain[35], we could use a similar animal holder with the ultrasonic arrays placed above the animal. Considering the multiple applications this system could have, we designed the entire system to be rotatable (Supplementary Fig. 11) such that it could fit various scenarios with different imaging subjects and positions. The clinical values of the system

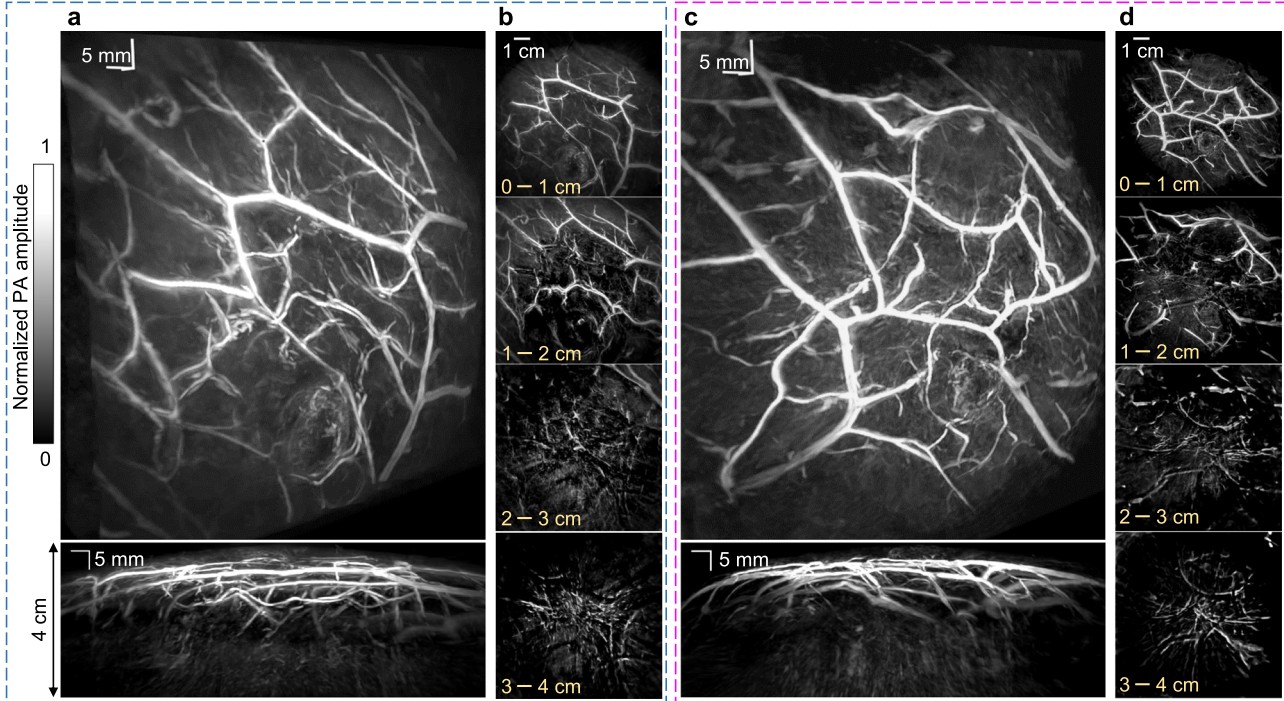

**Fig. 4 3D-PACT of human breasts in vivo. a** (top) Perspective angiogram of the right breast of a healthy human subject; (bottom) MAP image of the right breast view from the side. An imaging depth of 4 cm from the skin surface has been achieved (Supplementary Movie 3). **b** Cross-sectional images of the right breast on different coronal planes from the nipple to the chest wall. Each cross-sectional image is an MAP of a 1 cm-thick slice of the breast. **c**, **d** Left breast images of the same human subject (Supplementary Movie 4).

are highlighted by the collaborations it has brought to us with the physicians from the City of Hope National Medical Center (breast cancer), Cedars-Sinai Medical Center (plastic and reconstructive surgery), Rancho Los Amigos National Rehabilitation Center (neurological surgery), and Ronald Reagan UCLA Medical Center (neonatology).

In this pilot study, we demonstrated the versatility and scalability of the 3D-PACT system that has a high reliability in reproducibly generating high-quality images. This system has advanced state-of-the-art PA tomography by providing a 3D imaging modality with superior performance for both preclinical research and clinical practice. To promote this technology, we share several design principles, which proved to be highly useful: (1) Since PACT reconstruction assumes light has a uniform or predicable distribution, light illumination needs to be carefully designed. High optical fluence within the safety limit is preferable for high signal-to-noise ratio; (2) a large view angle of the transducer array or detection aperture is helpful to minimize the loss of information and improve the image quality; (3) since PA signals generated from the deep tissue could suffer both light and acoustic attenuation, analog pre-amplifiers connected between the ultrasonic arrays and DAQs can amplify the weak PA signals before cable noise compromises the SNR; (4) in addition to the temporal Nyquist sampling law, one should also consider the spatial Nyquist sampling criterion[22]. Inadequate spatial sampling will result in limited FOV and undersampling artefacts; (5) high imaging speed is helpful to reduce the motion-induced artefacts. Although co-registration methods can partially mitigate the motion distortion, the non-rigidity of the biological tissue compromises the effectiveness of the co-registration; (6) proper grounding of the metallic housing of the transducer array, pre-amplifiers, and DAQs is important to reduce noise. The motors and their drivers are often well-shielded to eliminate electromagnetic interference; (7) finally, a reliable system with outstanding performance is always well-organized, where a concise

design and neat arrangement facilitate problem debugging and reduce the likelihood of system failure.

## Methods

**System construction.** The 3D-PACT system is mainly comprised of an ultrasonic detection module, pre-amplification and DAQ circuits, a rotation scanner, and an illumination laser. To achieve 3D panoramic ultrasonic detection, we employed four arc-shaped ultrasonic transducer arrays (Imasonic, Inc.; 2.25 MHz central frequency; 1024 elements) with a separation of 90 degrees that were mounted on a hemispherical surface (Fig. 1c). In each array, 256 wideband transducer elements were integrated on an 83-degree arc with a 130-mm radius. Each element had a flat-rectangular aperture (0.6 mm × 0.7 mm element size; 0.74 mm pitch). The hemispherical surface was constructed with white plastic such that back-scattered light from the tissue surface would be partially recycled and reflected to the tissue. The ultrasonic array housing was electrically grounded and was supported by eight metallic pillars that were fixed on an aluminum rotary board (Fig. 1b and Supplementary Fig. 2).

Above the rotary board, 32 sets of custom 32-channel pre-amplifiers (51 dB gain) were directly connected to the base of the ultrasonic arrays' housing, amplifying the PA signals prior to experiencing cable noise. The amplified PA signals were then transmitted to four DAQ modules via four customized phase plates (PhotoSound, Inc.). Each DAQ module had 256-channel analog-to-digital converters (PhotoSound, Inc., ADC256; 40 MHz maximum sampling rate; 12 bit dynamic range), streaming the digitized data to a solid-state drive via USB 3.0. The DAQ modules were operated using National Instruments LabVIEW 2018 (64-bit).

To coaxially scan the ultrasonic transducer arrays for 90 degrees, we mounted two tapered-roller bearings with four-inch shaft diameter beneath the rotary board to provide coaxial alignment and steady rotation. The coaxial alignment was confirmed by aligning three one-inch holes on the ultrasonic array housing, rotary board, and driven spur gear with a one inch-diameter post. The rotation was motorized by a stepper motor (NEMA 34, 8 V) coupled with a set of two spur gears (Designatronics, Inc., KSS2-20J12 and KSS2-120, gear ratio = 1:6). The angular rotation speeds of the arrays were $\frac{\pi}{4}$ rad/sec, $\frac{\pi}{10}$ rad/sec, and $\frac{\pi}{20}$ rad/sec for 0.5-Hz functional imaging, 5-second brain anatomy imaging, and single breath-hold breast imaging, respectively. According to the spatial Nyquist sampling criterion[22], the FOV in brain anatomy imaging should have a diameter of ~2 cm, within which the spatial resolution is nearly isotropic (Supplementary Fig. 5). For rat abdomen and human breast imaging, we scanned the arrays for 10 s to provide a larger FOV (~4 cm). The lateral resolution outside the FOV was enlarged in relation to the increase in distance from the center (i.e., scanning axis)[22]. To generate a larger FOV within the same scanning time, one could use a faster laser (e.g., LPY7875-20, 20 Hz, Litron Lasers Ltd.) for illumination.

In 3D-PACT, an image reconstruction assumes a consistent light distribution in the imaging subject while scanning the ultrasonic arrays (i.e., spatial sampling)[24]. To achieve this, we guided the laser beam through the three coaxially aligned holes to a lens tube glued on the ultrasonic array housing. The lens tube was made of acrylic plastic to minimize the PA signal generated from it. The beam was further expanded by two engineered diffusers (EDC-40, EDC-15, RPC Photonics Inc.) sealed in the lens tube. The beam diameter was expanded to ~8 cm for human breast imaging (Supplementary Fig. 3). For rat brain and abdomen imaging, we added a condenser lens (ACL25416U-B, Thorlabs Inc.) after the diffusers (Supplementary Fig. 6a) to shrink the beam to a diameter of ~5 cm. As the light is scattered soon after propagating into biological tissues, we estimated that the energy distribution was relatively uniform across regions of the same depth. The optical fluence on the tissue surface was limited by the American National Standards Institutes safety standards[23]. Given the low optical attenuation in biological tissues, we utilized 1064-nm light for illumination, which is the fundamental wavelength of the Nd:YAG laser (PRO-350-10, Quanta-Ray, 10-Hz pulse repetition rate, 8–12-ns pulse width). To minimize the optical and acoustic energy loss, we used deuterium oxide ($D_2O$, Isowater Corp.) to acoustically couple the holding cup (3–4 cm deep) with the ultrasonic arrays. The space between the biological tissue and holding cup was filled by a small amount of bathwater.

To synchronize the 3D-PACT system, the laser's external trigger was used to trigger the DAQ modules, the rotation drive, and the linear power supply (IHE5-18/OVP, 5VDC, 18 A, International Power Corporation Ltd) of the pre-amplifiers (Supplementary Fig. 2). The power supply remained on during the experiment, whereas the power transmission to pre-amplifiers was controlled by a relay which functioned as a switch. Three seconds prior to the data acquisition, the relay was closed to transmit DC power to the pre-amplifiers. As the pre-amplification circuits were purely analog with the use of solid-state electronics, they amplified PA signals stably during data acquisition. Because the data acquisition time was limited, heat accumulation in the circuits was negligible. The four DAQ modules ran in parallel from a trigger supplied to the master DAQ module. The signal, trigger, and power cables were coiled around the rotation axis to reduce their required space. To prevent cable breaking, the system was programmed for round-trip scanning using a graphical user interface (GUI) generated by MATLAB 2018.

**Geometric calibration of the ultrasonic arrays.** Owing to manufacturing defects, the geometric position of each ultrasonic transducer element required calibration. We imaged a point source placed at multiple positions within the FOV and iteratively corrected the geometric location of each transducer element. To emit strong point source PA waves, a point absorber (~100 μm) was glued to an optical fiber tip such that most of the light coupled into the fiber was absorbed. Because the arrays' housing was coaxially aligned and mounted, we assumed all the transducer elements share the same rotation axis.

The distance between the point source and each transducer element can be described by the following quadratic equations:

$$(x_m - x_n)^2 + (y_m - y_n)^2 + (z_m - z_n)^2 = (c \cdot t_{m,n})^2 \qquad (1)$$
$$m = 1, 2, \ldots, M; n = 1, 2, \ldots, N$$

Here, $M$ and $N$ are the numbers of point sources and elements, respectively; $(x_m, y_m, z_m)$ denotes the location of the $m$-th point source; $(x_n, y_n, z_n)$ represents the location of the $n$-th transducer element; $t_{m,n}$ is the acoustic propagation time from the $m$-th point source to the $n$-th element; and $c$ is the speed of sound in water. The center of the hemispherical detection matrix is chosen to be the origin.

We can measure $t_{m,n}$ precisely since the DAQ samples at 40 MHz. We have initial estimations of the unknown parameters $c$ from its relationship with the water temperature, $(x_m, y_m, z_m)$ from the reconstructed images, and $(x_n, y_n, z_n)$ from the manufacture's specifications. As our initial estimations should be close to the true values, Eq. (1) can be assumed to be locally linear and the Jacobi iteration[59] was applied to find the true values of $(x_n, y_n, z_n)$. In each iteration, we updated the parameters as follows:

(1) Speed of sound $c$:

$$c = \frac{1}{MN} \sum_{M=1}^{M} \sum_{n=1}^{N} \sqrt{(x_m - x_n)^2 + (y_m - y_n)^2 + (z_m - z_n)^2} / t_{m,n} \qquad (2)$$

(2) Point source location $(x_n, y_n, z_n)$:

$$2(x_n - x_{n-1})x_m + 2(y_n - y_{n-1})y_m + 2(z_n - z_{n-1})z_m$$
$$= \left(c \cdot t_{m,n-1}\right)^2 - \left(c \cdot t_{m,n}\right)^2 + x_n^2 + y_n^2 + z_n^2 - x_{n-1}^2 - y_{n-1}^2 - z_{n-1}^2 \qquad (3)$$
$$m = 1, 2, \ldots, M; n = 2, 3, \ldots, N$$

As $M$ was much less than $N$, we applied the method of least squares method to solve the linear system and obtained $(x_m, y_m, z_m)$.

(3) Element location $(x_n, y_n, z_n)$

Because the ultrasonic arrays were manufactured by saw dicing[60], the geometric displacement along the lateral direction is negligible. Accordingly, we only calibrated the geometric defects in the radial direction. We defined a radial calibration factor $\alpha_n$ to confine the geometric defects along the radial axis and

solved it from the quadratic equation:

$$\sum_{m=1}^{M} (x_m - \alpha_n x_n)^2 + (y_m - \alpha_n y_n)^2 + (z_m - \alpha_n z_n)^2 = \sum_{m=1}^{M} \left(c \cdot t_{m,n}\right)^2 \qquad (4)$$

We iteratively updated the three sets of parameters until they converged (50 times iteration). Supplementary Fig. 4a shows the images of a leaf skeleton reconstructed before the geometric calibration. Supplementary Fig. 4b shows the radial calibration factor $\alpha_n$ of each transducer element. The image quality is clearly improved after calibration (Fig. 1d).

**3D image reconstruction.** We used the universal back-projection (UBP) algorithm[24] (Supplementary Methods) implemented in C++ to reconstruct all images in this work. The time-domain PA signals acquired at all scanning steps were back-projected into the 3D space using a GPU-accelerated implementation of the UBP. To mitigate aliasing artefacts caused by spatial undersampling in regions outside of the well-resolved FOV, we applied spatial interpolation and low-pass filtered PA signals with cutoff frequencies determined by the distance to the center of the imaging aperture[22,61]. A dual-speed-of-sound UBP algorithm[61] was then used to mitigate the artefacts induced by the acoustic inhomogeneity between the biological tissue and the coupling fluid ($D_2O$). We first reconstructed an initial image using the conventional UBP with a single speed sound[24], then extracted the biological tissue's surface (i.e., holding cup's surface) from the initial image by moving detection windows towards the biological tissue and identifying the location of the first significant increase in PA amplitudes. The extracted surface was further fitted to a half ellipsoid. We then applied the dual-speed-of-sound UBP algorithm by assigning the speed of sound in tissue and $D_2O$ to regions inside and outside the ellipsoid, respectively.

Each volumetric image was reconstructed with a voxel size of $0.13 \times 0.13 \times 0.13$ mm$^3$. We further batch-processed all the reconstructed images to improve contrast. For image post-processing, we first applied a depth compensation ($e^{0.81 \times depth(cm)}$ for brain; $e^{0.72 \times depth(cm)}$ for breast) method to enhance the PA amplitude in the deep tissue. The compensated images were then denoised using sparse 4D transform-domain collaborative filtration[62]. We further applied Hessian-based Frangi vesselness filtration[63] to the denoised images to enhance the contrast of blood vessels. Finally, we added the filtered images (self-normalized) with a weighting factor of 0.2 back to the original depth-compensated images (with a weighting factor of 0.8) and obtained the presented images. The 3D-rendered rat brain image (Supplementary Movie 1) was generated using VolView 3.4, and the 3D-rendered breast images (Supplementary Movies 3 and 4) were produced using the visvis library in Python 3.7.4.

**Time-gated motion correction.** Because the rat brain and human breast were well-fixed during imaging, motion correction was not applied to reconstruct the presented images. However, artefacts induced by periodic motions can be largely removed by time-gated motion correction (Supplementary Fig. 9). For example, when we scanned the abdominal region of a 3-month-old pregnant rat (Hsd: Sprague Dawley SD, Harlan Co.; Timed mated, 20 days) using 3D-PACT, respiration-induced motions appeared as spikes in the acquired PA signals (Supplementary Fig. 9a). To recover the signals acquired during motion, we detected the first-arrival times of the PA signals in different channels and applied a median filter with a kernel size of three channels to the first-arrival times. We then calculated the numerical difference of the signals' first-arrival times. Signals with numerical differences >0.025 μs (temporal sampling step size) were regarded as motion-affected signals and were removed. The remaining signals formed a motionless partial-scan detection. Since the breathing cycles of the subject (i.e., rat abdomen) were spatially repeatable, we repeated the above procedures with four sequential 10-second scans and spliced the motionless partial-scan signals to form a full-scan detection (Supplementary Fig. 9a). The motionless full-scan signals were then used to reconstruct the image, in which the PA signals better converged (Supplementary Fig. 9b). Equipped with a faster laser, we could use this time-gated motion correction method to image faster phenomena (e.g., heartbeats).

**Animal preparation.** Adult, 1-month-old rats (Hsd:Sprague Dawley SD, Harlan Co.; 100–120 g body weight, male) were used for functional brain imaging. Before imaging, the rat was maintained under anesthesia with 1.5% vaporized isoflurane and its eyes were covered by pharmaceutical-grade ophthalmic ointment. The hair on the head was first removed using clippers and depilatory cream. We then opened the scalp, thinned the cranial surface with a handheld drill, and polished the surface with a green stone burr. For functional imaging, the skull was kept intact but the top part of the scalp was removed. Afterwards, we placed a 3D-printed nose cone and tooth bar on the rat and sealed the gap with silicone glue to prevent leakage.

The rat was then placed in a supine position and secured to a lab-made animal holder that was covered by a feedback-controlled heating pad to maintain the animal's body temperature. To measure the brain's response to hypoxic challenges, we switched the inhaled gas between air and pure nitrogen (Fig. 3b, c). To monitor the hemodynamics while awakening from deep anesthesia, we turned off the isoflurane vaporizer until the rat started to slightly move its limbs. During the measurement of intrinsic function connectivity and hemodynamics in response to the electrical stimulations, the isoflurane level was maintained at 0.6% so that the

brain could be more active. The electrical stimulations were applied to the rat by binding two electrodes from an isolated pulse stimulator (A-M Systems, Inc., Model 2100) to one of the rat's front limbs. The stimulator delivered 2-mA pulsed current to the electrodes with a pulse width of one millisecond. Each stimulation period lasted 12 s (4 Hz pulse frequency), followed by a 12-second rest (Fig. 3h).

**Intrinsic functional connectivity of the rat brain**. We adopted a data analysis method previously reported[64] to measure the intrinsic functional connectivity of the rat brain. Each functional connectivity measurement took 8 min, with a 0.5 Hz volumetric imaging rate of the whole rat brain. After reconstructing all the 240 ($0.5 \times 60 \times 8 = 240$) time sequential images, a second-order Butterworth bandpass filter (0.008–0.09 Hz) was first applied to all temporal sequences, followed by a global signal regression based on the time sequences of all pixels within the brain. For functional connectivity analyses, we identified 20 functional regions throughout the brain, averaged the signals from pixels within each region, and computed correlation coefficients between regions to form a connectivity network matrix (Fig. 3g). Supplementary Movie 2 was made using a specialized 3D visualization software package, namely 3D PA Visualization Studio[65].

**Imaging protocols**. All human and animal imaging experiments were performed in accordance with relevant guidelines and regulations. The human imaging experiment followed the protocol approved by the Institutional Review Board (IRB) of the California Institute of Technology. Human breast imaging was performed in a dedicated imaging room. After verbally agreeing to participate in the study, the subject signed the informed consent form, which was locked in a cabinet by researchers. The animal experiments followed the protocol approved by the Institutional Animal Care and Use Committee (IACUC) of the California Institute of Technology. IRB and IACUC were aware of both protocols before approving.

**Reporting summary**. Further information on research design is available in the Nature Research Reporting Summary linked to this article.

## Data availability

All data are available within the Article and Supplementary Files, or available from the corresponding author upon reasonable request. Imaging data from in vivo experiments and study approvals issued by the IRB and IACUC are available in the online folder: https://figshare.com/articles/dataset/3D-PACT_Data_and_Codes/13114544.

## Code availability

The image reconstruction and processing algorithms are described in detail in Methods. The reconstruction code is not publicly available because the code is proprietary and is used in licensed technologies. Pseudo-code and description of the universal back-projection algorithm used in this study are provided in the Supplementary Methods. Imaging processing and analyzing programs are available in the corresponding data folders.

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

## Acknowledgements

We thank Dr. Vassili Ivanov for useful discussions. This work was sponsored by the United States National Institutes of Health (NIH) grants R01 CA186567 (NIH Director's Transformative Research Award), R35 CA220436 (Outstanding Investigator Award), and U01 NS099717 (BRAIN2 Initiative).

## Author contributions

L.V.W. and L.L. conceived and designed the study. L.L., D.C.G., J.S. and K.M. constructed the hardware system. S.N., L.L. and X.Y. developed the control program. L.L. and X.T. performed the experiments. R.C. designed the functional rat brain imaging experiments and performed the skull-thinning surgery. P.H. developed the reconstruction algorithm. L.L., X.T. and P.H. analyzed the data. L.V.W. supervised the study. All authors wrote the manuscript.

## Competing interests

All the authors declare no competing interests. K.M. has a financial interest in Micro-Photoacoustics, Inc. L.V.W. has a financial interest in Microphotoacoustics, Inc., Cal-PACT, LLC, and Union Photoacoustic Technologies, Ltd., which, however, did not support this work.
