## [Peer Review File · Nature Communications]

REVIEWER COMMENTS

Reviewer #1 (Remarks to the Author):

In this paper, the authors developed a three-dimensional photoacoustic computed tomography (3D-PACT) system. It achieved fast, large depth, adjustable FOV, multi-purpose photoacoustic imaging capability. With a FOV of a human breast, it still offered sub-millimeter isotropic resolution for visualizing the vasculature, which is very impressive. Other experiments demonstrated that this system was able to offer structural and functional imaging of the entire mouse brain. Overall speaking, this paper holds good novelty for PACT system development. However, authors still need to address some of my concerns before acceptance.

-Major concerns:

- 1) Authors claim that two engineered diffusers are used to expand the beam for uniform illumination. This is very important for achieving isotropic resolution. Therefore, it is necessary for authors to conduct another experiment to measure how uniform the excitation light actually is, especially for different FOVs.
- 2) There is a mechanical scanning module that rotates the transducer arrays coaxially for 90 degrees. How fast it is? Because the volumetric imaging rate is 0.5 Hz. The rotation speed must be fast enough to meet this performance.
- 3) This system successfully acquired images of a mouse brain, a pregnant rat abdomen, and human breasts. They all have different FOVs. Were the spatial resolutions and other important specifications all consistent in these FOVs. Were hardware setups all the same for different FOVs? Or there is some minor changes of the setup for imaging different FOVs? Please clarify these details.
- 4) In the discussion section, authors should also list some limitations of this system, even though it is, indeed, a very impressive PACT system. For example, lacking multispectral capability, very fast but still not true real-time, not an ideal wavelength for imaging clinically approved ICG.

-Minor concerns:

- 1) There is a messy code in Fig. 1a. please modify the 4□256-channel DAQs to 4x256-channel DAQs.

2) It is better to label some important specifications on the picture of the 3D-PACT system, such as the diameter of the imaging aperture, the depth of the holding cup, the diameter of the diffuser, the cover area of the diffused excitation light, etc.

Reviewer #2 (Remarks to the Author):

Lin and colleagues describe a photoacoustic computed tomography system that allows imaging several centimeters into tissue with a large field of view, using hemoglobin as an endogenous contrast agent. The system is used to image the rat brain, rat fetuses, and the human breast. The paper is clearly written, and the data is of high quality. However, in this reviewer's eyes the technical advance, particularly the rodent data, is limited when compared with current methods.

The system is used to measure functional activation and resting state connectivity in rats. One stated advantage of the method is its higher speed relative to other more established methods (fMRI). However, neither of these experiments make use of the speed (functional activation is relatively slow on the order of a few seconds, functional connectivity looks at frequencies <0.1 Hz), so the advantage of this technique over fMRI is minimal. Furthermore, the animals must be anesthetized and inverted for the imaging. Many groups have moved to using awake rodents in fMRI (Ferenczi, Science 2016 doi:10.1126/science.aac9698), which I don't think this system can do. There are potentially interesting scientific questions that the higher spatial resolution can answer that fMRI cannot (for example, what sort of dynamics do the large arteries and veins have during the resting state?), but these issues were not attacked.

The change in signal seen when awakening from anesthesia (figure 3d) is ascribed to changes in oxygenation. This is most likely not the case, but rather the removal of isoflurane, which is has been known for decades as a very powerful vasodilator (see for example Archer et al. Anesthesiology. 1987;67(5):642-8. doi: 10.1097/00000542-198711000-00005.)

Minor concerns

Line 310 "...non-rigidity of the biological tissue compromises the effectiveness" should have "of the co-registration" at the end

Manufacturing "errors" : I think "defect" is a more appropriate term here

Line 355: "...the pre-amplifiers were only powered during scanning..." Did the warm-up time of the amplifiers affect the stability of the signal?

Reviewer #3 (Remarks to the Author):

Authors present a detailed report of a high-speed 3D photoacoustic Computed Tomography system and demonstrate its working on the breast of a human volunteer and the rat brain. The device is a sophisticated system and the authors cover its engineering aspects in a very detailed manner. They also present visualizations of pleasing images and have extracted in the animal study physiological parameters which may be important in arriving at clinical decisions. While this is a valuable paper for a specialist journal, it falls short of making the grade to Nature Communications. The reasons for this are the following:

- 1) The advancement in the science is limited, while the engineering is definitely of very high class
- 2) While the engineering is of high class, the system does not represent a breakthrough as it is presented.
- 3) Authors have not consulted the literature on state-of-the-art systems
- 4) The contributions are not placed in the clinical context of for example breast cancer by showing images of breast lesions.

Here some other points while re-submitting:

i) Line 89: What is the meaning of 'clarity'? Further the word 'quality' of image is used often. Define this and make it measurable.

ii) From line 59: There are other spherical geometries recently published: Oraevsky et al (2019) in Photoacoustics, Shoustra et al (2019) in JBO.

iii) Line 47: I do not think that Optical tomography can be described as a mainstream imaging modality.

iv) Penetration depth of 4 cm and resolution of around 0.4 mm is very impressive.

v) Line 254: It is not clear why this statement especially about a certain value of absorption coefficient has clinical significance: The system is sufficiently sensitive to detect a 1 mm-diameter tumor with an absorption coefficient 2.1 times as high as the surrounding phantom.

vi) Materials and Methods are written very elaborately. These are very useful for the interested reader in a specialist journal where it should be published.

Reviewers' Comments and Authors' Response

Reviewer #1:

In this paper, the authors developed a three-dimensional photoacoustic computed tomography (3D-PACT) system. It achieved fast, large depth, adjustable FOV, multi-purpose photoacoustic imaging capability. With a FOV of a human breast, it still offered sub-millimeter isotropic resolution for visualizing the vasculature, which is very impressive. Other experiments demonstrated that this system was able to offer structural and functional imaging of the entire mouse brain. Overall speaking, this paper holds good novelty for PACT system development. However, authors still need to address some of my concerns before acceptance.

Major concerns:

1) Authors claim that two engineered diffusers are used to expand the beam for uniform illumination. This is very important for achieving isotropic resolution. Therefore, it is necessary for authors to conduct another experiment to measure how uniform the excitation light actually is, especially for different FOVs.

Reply: Thank you for the constructive suggestion. The main purpose of applying two diffusers (EDC-40, EDC-15, RPC Photonics Inc.) is to ensure safety; the laser beam would be expanded even if one of the diffusers was broken or misaligned during imaging. We measured the light intensity distribution for different FOVs (Supplementary Fig. 3). The beam diameter was expanded to ~8 cm for human breast imaging. For rat brain and abdomen imaging, we added a lens (ACL25416U-B, Thorlabs Inc., immersed in the D₂O) after the diffusers to shrink the beam to a diameter of ~5 cm. We added light intensity distribution (Supplementary Fig. 3) to the Supplementary Information and a detailed description of the optical components to the manuscript on Page 18. In addition, the estimated optical speckle size on the tissue surface is around 10 μm [1], and the speckle grain count within a voxel is $(\frac{130 \mu m}{10 \mu m})^2$. Accordingly, the optical heterogeneity in each voxel is reduced by 13X. Since the light is scattered soon after propagating into biological tissues, we estimated the energy distribution was relatively uniform across regions of the same depth.

Optical illumination does not affect the acoustically-defined spatial resolution, but the energy distribution of the excitation light affects the uniformity of system's sensitivity (i.e., the ability to detect weak signals from small features) within the FOV.

2) This system successfully acquired images of a mouse brain, a pregnant rat abdomen, and human breasts. They all have different FOVs. Were the spatial resolutions and other important specifications all

consistent in these FOVs. Were hardware setups all the same for different FOVs? Or are there some minor changes of the setup for imaging different FOVs? Please clarify these details.

Reply: Thank you for the helpful suggestion. The isotropic spatial resolution was acoustically determined upon the satisfaction of the spatial Nyquist sampling criterion [2]. For example, we scanned the 4×256-element arrays over 90 degrees in 5 seconds to image a rat brain, providing ($5 \text{ s} \times 10 \text{ Hz} \times 4 \text{ arrays} = 200$) lateral scanning steps (i.e., sampling points). According to the spatial Nyquist sampling criterion, the FOV should have a diameter of $\sim 2 \text{ cm}$, within which the spatial resolution is nearly isotropic (Supplementary Fig. 5).

For human breast imaging, we scanned the arrays during a single breath hold of 10 seconds to provide a larger FOV ($\sim 4 \text{ cm}$). The lateral resolution outside the FOV was enlarged in relation to the increase in distance from the center (i.e., scanning axis) [2]. To mitigate aliasing artefacts caused by spatial under-sampling in regions outside of the FOV, we applied spatial interpolation and low-pass filtered PA signals with cut-off frequencies determined by the distance to the center of the imaging aperture [2]. To generate a larger FOV within the same scanning time, one could also use a faster laser (e.g., LPY7875-20, 20 Hz, Litron Lasers Ltd.) for illumination, though the safety limit of the optical fluence on the skin surface would be half of that for 10-Hz lasers [3].

For rat abdomen imaging, we repeated four sequential 10-second scans and spliced the motionless partial-scan signals to form a full-scan detection. The FOV is the same as that in breast imaging.

Therefore, for different imaging applications, the setup changes mainly related to the optical components (with or without a condenser lens) and scanning speed. For human breast imaging, we also placed a soft imaging bed above the system to provide the subject with comfortable positioning. The imaging room setup is shown in a virtual tour (video link: <https://caltech.app.box.com/s/afdlf12barmgq8zxs372733wlrqfve8f>).

The discussions above have been included in the manuscript on Pages 18 and 21.

3) There is a mechanical scanning module that rotates the transducer arrays coaxially for 90 degrees. How fast it is? Because the volumetric imaging rate is 0.5 Hz. The rotation speed must be fast enough to meet this performance.

Reply: Since we have four arc-shaped ultrasonic arrays with 90-degree separation, we scanned the arrays over 90 degrees to form a hemispherical detection surface. For 0.5-Hz functional imaging, the angular rotation speed of the arrays was $\omega = \frac{\pi}{4} \text{ rad/sec}$. The rotation was driven by a stepper motor (NEMA 34, 8V) coupled with a set of two spur gears (gear ratio = 1:6). Therefore, the motor rotated at $\omega = \frac{3\pi}{2} \text{ rad/sec}$. The imaging speed could be further increased by using a faster laser (e.g., LPY742-100, 100 Hz, Litron

Lasers Ltd.) and changing the stepper motor to a servomotor with higher torque (e.g., EMG-50ASA, Anaheim Automation Inc.).

We added the above information to the manuscript on Pages 16 and 18.

4) In the discussion section, authors should also list some limitations of this system, even though it is, indeed, a very impressive PACT system. For example, lacking multispectral capability, very fast but still not true real-time, not an ideal wavelength for imaging clinically approved ICG.

Reply: Thank you for the compliment. We included the following discussion to the manuscript on Page 15.

The current imaging system has limitations in its single-wavelength illumination and mechanical scanning time (2–10 seconds). To further improve the performance of 3D-PACT, several upgrades could be made by (1) using multi-wavelength illumination to provide additional imaging contrast (e.g., indocyanine green); (2) using lasers with higher pulse energy to create a larger illumination area for breast imaging; (3) using lasers with higher repetition rates to image a larger FOV during the same scanning time; (4) replacing the stepper motor to a servomotor with higher torque to allow faster mechanical scanning. Real-time 3D imaging is potentially achievable by employing more array elements for parallel detection and sparse-sampling algorithms for reconstruction [4,5].

Minor concerns:

5) There is a messy code in Fig, 1a. please modify the 4□256-channel DAQs to 4x256-channel DAQs.

Reply: Thank you for the catching the messy code. It was caused by the format transition from Word to PDF. We have corrected this error.

6) It is better to label some important specifications on the picture of the 3D-PACT system, such as the diameter of the imaging aperture, the depth of the holding cup, the diameter of the diffuser, the cover area of the diffused excitation light, etc.

Reply: Thank you for the constructive suggestion. We have added the system specifications to Supplementary Fig. 6.

Reviewer #2:

Lin and colleagues describe a photoacoustic computed tomography system that allows imaging several centimeters into tissue with a large field of view, using hemoglobin as an endogenous contrast agent. The system is used to image the rat brain, rat fetuses, and the human breast. The paper is clearly written, and the data is of high quality. However, in this reviewer's eyes the technical advance, particularly the rodent data, is limited when compared with current methods.

Major concerns:

1) The system is used to measure functional activation and resting state connectivity in rats. One stated advantage of the method is its higher speed relative to other more established methods (fMRI). However, neither of these experiments make use of the speed (functional activation is relatively slow on the order of a few seconds, functional connectivity looks at frequencies <0.1 Hz), so the advantage of this technique over fMRI is minimal. Furthermore, the animals must be anesthetized and inverted for the imaging. Many groups have moved to using awake rodents in fMRI (Ferenczi, Science 2016 doi:10.1126/science.aac9698), which I don't think this system can do. There are potentially interesting scientific questions that the higher spatial resolution can answer that fMRI cannot (for example, what sort of dynamics do the large arteries and veins have during the resting state?), but these issues were not attacked.

Reply: Thank you for the helpful suggestion. To demonstrate the advantage of 3D-PACT over fMRI in spatial resolution, we further quantified the hemodynamics in the major arteries and veins in the rat brain (Fig. 3c, f, and Supplementary Fig. 8) during hypoxic challenges and the awakening process. At 1064 nm, oxyhemoglobin dominates the optical absorption (i.e., $\mu_{a(HbO_2)} \approx 10 \times \mu_{a(Hb)}$) [6]. We selected regions of interest (ROI) slightly larger than the blood vessels' cross-sections and monitored the changes in spatially-averaged PA signals within the ROI over time. Therefore, both the oxygen saturation (sO_2) and the blood volume (or vascular diameter) within the ROI affect PA signal amplitudes.

During hypoxic challenges, we detected similar hemodynamics in four major arteries (azygos of the anterior cerebral artery, AACA; olfactory artery, OA) and veins (superior sagittal sinus, SSS; inferior sagittal sinus, ISS) (Fig. 3c and Supplementary Fig. 8a) [7]. We also noticed that the PA signals decreased more in the four major blood vessels (Fig. 3c) compared to in brain tissue (Fig. 3b), which includes both blood vessels and brain cells. To explain this difference in PA signals' change, we surmise that in Fig. 3b, the time-invariant PA signals from lipid, water, and proteins [8] moderated the change in PA signals from hemoglobin.

Another observation is that the slopes of the PA signals plateaued during the 40-second hypoxia in Fig. 3b and 3c, which is likely due to a reduced oxygen consumption rate in response to extended hypoxia.

In comparison, the slopes of the PA signals are more consistent during the 10-second hypoxia. This agrees with the expectation that oxygen consumption rate is more stable at the beginning of hypoxia [9].

Moreover, we also measured hemodynamics in the same major blood vessels during the awakening process (Fig. 3f and Supplementary Fig. 8b). Similar hemodynamics were observed in the two arteries and two veins. To interpret this similarity, we surmise that in veins, both the decrease in sO_2 and the contraction of vascular diameter [10] contribute to the PA signals' decrease. In arteries, although sO_2 is expected to be more stable during the awakening process [10], the vascular contraction may be more pronounced than that in veins [10].

Another observation is that the decrease in PA signals from major vessels (Fig. 3f) was comparable to that in brain tissues (Fig. 3e). To explain this, our hypothesis is that although the time-invariant PA signals from lipid, water, and proteins could moderate the changes in PA signals from hemoglobin, small vessels in the brain tissue are expected to contract more than major vessels during the awakening process [10], thus inducing a decrease in PA signals as seen in Fig. 3e.

The spatial resolution of 3D-PACT could be further improved by replacing the ultrasonic arrays with higher center frequencies (i.e., wider bandwidth). We added the above data analysis to the manuscript on Pages 8, 9, 10 and 11.

Although the imaging speed of 3D-PACT for human breast imaging (10 seconds) is much faster than breast MRI, the superior imaging speed for small animal functional imaging is not yet obvious compared with fMRI. The imaging speed in our current system is mainly limited by the laser's repetition rate (10 Hz) and the torque of the stepper motor (1288 oz-in). With the current hardware configuration, however, the spatial sampling would be too sparse [2] and the motor generated vibration when the system scanned faster than 0.5 Hz. By using a faster laser (e.g., LPY742-100, 100 Hz repetition rate, Litron Lasers Ltd.) and a servomotor with higher torque (e.g., EMG-50ASA, 3384 oz-in torque, Anaheim Automation Inc.), the system could potentially scan at 5 Hz for functional 3D imaging. The imaging speed could be further improved by adding more ultrasound detection channels.

We included the discussion above to the manuscript on Pages 12 and 16 and revised the previous statement regarding the advantage in imaging speed over fMRI. In summary, for small animal functional imaging, while the imaging speed of 3D-PACT is not yet substantially faster than fMRI, the high spatial resolution of 3D-PACT enables measurement of hemodynamics in individual vessels. To improve the imaging speed, one could employ a laser with higher repetition rate and a motor with higher torque to provide dense sampling and fast scanning.

In addition to the high spatial resolution demonstrated in the manuscript, we summarized several other benefits of PACT over fMRI for small animal functional imaging (Table 1). The optical absorption

contrast enables PACT for functional and molecular imaging with either endogenous contrast (e.g., blood volume, oxy- and deoxyhemoglobin) or exogenous contrast (e.g., organic dyes, reporter gene products). Multi-wavelength illumination would express wavelength-dependent light absorption and exploit this strength to the fullest. Furthermore, PACT images optical absorption with 100% sensitivity because non-absorbing components present no background. In contrast, BOLD fMRI detects neural activities based on only deoxyhemoglobin in a nonlinear relationship with a modest sensitivity due to substantial tissue background. Although some of the MRI contrast agents have been applied in animals, only Mn^{2+} dependent labeling has been used for functional imaging of neural activity [11]. Free of strong magnetic fields, PACT is able to obtain functional images in subjects with ferromagnetic implants/devices, which preclude imaging with MRI. For the same reason, PACT will be favorable for the introduction of neuromodulators and complementary imaging/recording modalities, in addition to its being potentially portable, relatively quiet during operation, and much less expensive.

Table 1. Strengths of PACT over fMRI for small animal imaging. BG, Background; BV, blood volume; HbO₂, oxyhemoglobin; HbR, deoxyhemoglobin.

	Endogenous contrast	BG	Sensitivity	Linearity	Portability	Platform	Sound	Magnet	Cost
fMRI	HbR	High	Low	No	No	Closed	Noisy	Yes	High
PACT	HbO ₂ , HbR, BV	Low	High	Yes	Yes	Open	Quiet	No	Low

It is likely feasible to image awake rodents using 3D-PACT, but the system would need major modifications and thus may no longer be suitable for human breast imaging. Inspired by the photoacoustic microscopy of the awake mouse brain [10], we could use a similar animal holder with the ultrasonic arrays placed inversely above the animal. Negative pressure would be needed to reduce the water pressure on the animal. Ultrasonic arrays with smaller footprints (i.e., diameter) would be ideal for easy operation. All the other components (e.g., light delivery, electronics, mechanics, control programs, reconstruction algorithms) could remain the same. The discussion above has been included in the manuscript on Page 16.

2) The change in signal seen when awakening from anesthesia (figure 3d) is ascribed to changes in oxygenation. This is most likely not the case, but rather the removal of isoflurane, which is has been known for decades as a very powerful vasodilator (see for example Archer et al. Anesthesiology. 1987;67(5):642-8. doi: 10.1097/00000542-198711000-00005.)

Reply: Thank you for the correction. We revised our explanation on Page 11 as follows:

Isoflurane, a well-known vasodilator [12], can also reduce the cerebral oxygen extraction fraction [10, 13]. Therefore, the decrease in PA signals within different brain regions from deep to light anesthesia (Fig. 3 d, e) could be a combined effect of reduced blood volume and oxygen saturation.

Our measurements in Fig. 3f also support this explanation.

Minor concerns:

3) Line 310 "...non-rigidity of the biological tissue compromises the effectiveness" should have "of the co-registration" at the end.

Reply: Thank you. We revised the manuscript as suggested on Page 17.

4) Manufacturing "errors" : I think "defect" is a more appropriate term here.

Reply: Thank you. We revised the manuscript as suggested.

5) Line 355: "...the pre-amplifiers were only powered during scanning..." Did the warm-up time of the amplifiers affect the stability of the signal?

Reply: Thank you. We have taken this into consideration when building the system. The linear power supply (IHE5-18/OVP, 5VDC, 18A, International Power Corporation Ltd) was remained on during the experiment, while the power transmission to preamplifiers was controlled by a relay which functioned as a switch. Three seconds prior to the data acquisition, the relay was closed to transmit DC power to the preamplifiers. Since the preamplification circuits were purely analog with the use of solid-state electronics, they amplified PA signals stably during data acquisition. Because the data acquisition time was limited, heat accumulation in the circuits was negligible.

We added the above information to the manuscript on Page 19.

Reviewer #3:

Authors present a detailed report of a high-speed 3D photoacoustic Computed Tomography system and demonstrate its working on the breast of a human volunteer and the rat brain. The device is a sophisticated system and the authors cover its engineering aspects in a very detailed manner. They also present visualizations of pleasing images and have extracted in the animal study physiological parameters which may be important in arriving at clinical decisions. While this is a valuable paper for a specialist journal, it falls short of making the grade to Nature Communications. The reasons for this are the following:

1) The advancement in the science is limited, while the engineering is definitely of very high class.

Reply: Thank you for the compliment from the engineering perspective. According to the scope of Nature Communications, which is a multidisciplinary journal dedicated to publishing high-quality research in all areas, we hope our exciting work could fit the journal well and provide a new imaging tool with irreplaceable advantages for preclinical research and clinical translation. The device we built is the first photoacoustic imaging system that can penetrate the whole rat brain and human breast with high isotropic spatial resolution as well as the high imaging speed. This breakthrough (as shown in Table 2) has already excited both preclinical researchers (e.g., Prof. Tzung Hsiai at UCLA) and clinical practitioners (see Discussion) seeking our collaboration. It is expected that the imaging capabilities introduced by this system may be of great interest to readers of Nature Communications for approaching and enabling other scientific pursuits. In addition, the work helps to establish standards for photoacoustic computed tomography (PACT).

2) While the engineering is of high class, the system does not represent a breakthrough as it is presented.

Reply: To the best of our knowledge, there are no other photoacoustic imaging systems which can provide comparable performance, including the penetration depth, isotropic spatial resolution, scalable FOV, and imaging speed (Table 2). Our work, representing the cutting-edge performance with high versatility, has recently been utilized for non-invasive heart imaging. The work has started to be expanded and more follow-up studies will be performed by other photoacoustic imaging groups. In addition, the work helps to establish standards for PACT including system design, construction, and image processing that will benefit the whole field.

Table 2 compares our 3D-PACT and the state-of-the-art PACT systems capable of 3D imaging. For small animal brain imaging, the Zurich-PACT [14] sacrificed the image quality for high imaging speed. For breast imaging, all the other state-of-the-art PACT systems [15-17] suffered from limited imaging

depth due to either sparse spatial sampling or suboptimal illumination. The side-by-side comparison in Table 2 clearly shows that our 3D-PACT has made a quantum leap forward.

Table 2. Comparison of 3D-PACT and other state-of-the-art PACT systems capable of imaging the small animal brain or human breast in 3D space.

	In vivo imaging depth	Imaging speed and well-resolved FOV	Functional imaging	Small animal brain imaging	Human breast imaging
Caltech 3D-PACT	10 mm in rat brains; 40 mm in human breasts	0.5-Hz volumetric rate for functional imaging (can be increased to 5 Hz or higher); FOV = 2.2 cm ³ 10 seconds for 3D human breast imaging FOV = 50.2 cm ³	Hemodynamics during 1) Hypoxic challenges; 2) Deep to light anesthesia; 3) Resting state 4) Electrical stimulations on limbs		Zurich PACT [14]	7 mm in mouse brains	20–25 Hz for mouse brain functional imaging FOV = 0.033 cm ³	1) Hemodynamics and 2) GCaMP6 responses during electrical stimulations in hind paws		Not demonstrated
Canon PACT [15]	~10 mm in human breasts	2 minutes for 3D human breast imaging FOV = 46.2 cm ³	S-factor (i.e., estimated oxygen saturation)	Not demonstrated	Tomo-Wave PACT [16]	Not demonstrated in vivo	10.6 minutes for 3D human breast imaging FOV = 2.1 cm ³	Oxygen saturation	Not demonstrated	
Twente PACT [17]	22 mm in human breasts (not shown explicitly)	4 minutes for 3D human breast imaging FOV = 6.2 cm ³	Not demonstrated	Not demonstrated	----------------------------	---	--	------------------	------------------	---

3) Authors have not consulted the literature on state-of-the-art systems.

Reply: The systems consulted in the Introduction, though they may not be the newest, represent the best performance of 3D PACT. To give a more comprehensive comparison, we discussed two more recent publications in the Introduction on Page 4:

More recent 3D PACT systems built by Oraevsky et al. [16] and Schoustra et al. [17] employ scanned arc ultrasonic arrays for human breast imaging. Both studies showed neither cross-sectional breast images at different depths nor 3D-rendered views, presumably due to the limited imaging depth. The system built by Oraevsky et al. [16] scanned an ultrasonic array with 96 elements over 320 rotational steps. According to the Nyquist sampling criterion [2], such a sparse spatial sampling would provide a well-resolved FOV with a diameter around 2 cm. Accordingly, the image quality was compromised while also requiring a long scanning time of several minutes. Similarly, the system built by Schoustra et al. [17] has a similar problem with limited FOV due to the sparse spatial sampling. In addition, the scanning of the laser beam varied the light energy distribution, which violates the assumption of 3D image reconstruction algorithms that light energy distribution remains consistent during scanning [18, 19].

The use of arc ultrasonic arrays in 3D-PACT should not equal to lack of novelty. In our system, careful design of the detection matrix, optimization of illumination, the use of pre-amplifiers, the practice of grounding and shielding, the correction of manufacturing defects, the reliable imaging strategy, and the improved 3D image reconstruction/processing algorithms all contribute to the unparalleled performance.

4) The contributions are not placed in the clinical context of for example breast cancer by showing images of breast lesions.

Reply: Breast cancer imaging has already been demonstrated by a 2D-version PACT system we published in Nature Communications in 2018 [20]. The 3D-PACT, equipped with isotropic spatial resolution and higher imaging speed, is expected to reveal breast cancer as well. The clinical study was temporally closed at Caltech to protect patients from unnecessary exposure during the coronavirus pandemic. Nonetheless, the main purpose of this manuscript is not to validate a specific application, but to present a world-leading 3D PACT system with high performance and versatility which will enable new preclinical research and clinical translation.

Here some other points while re-submitting:

5) Line 89: What is the meaning of 'clarity'? Further the word 'quality' of image is used often. Define this and make it measurable.

Reply: Thank you for the constructive suggestion. We defined clarity as sufficient contrast-to-noise ratio (CNR) to reveal detailed structures on the order of the inherent spatial resolution. A useful rule of thumb is that a CNR of 2.2 can be detected with a confidence level of 98% [21]. At a depth of 1 cm in the rat brain and 4 cm in the human breast, the CNRs of ~0.4 mm vessels are 3.7 and 2.8, respectively. We defined high image quality as the ability to provide clarity within an FOV large and deep enough to cover the target region.

We added the above definitions to the manuscript on Page 3.

ii) From line 59: There are other spherical geometries recently published: Oraevsky et al (2019) in Photoacoustics, Schoustra et al (2019) in JBO.

Reply: Thank you for the helpful suggestion. We included the discussion about the two papers in the manuscript on Page 4. We also compared the performances of the state-of-the-art PACT systems in Table 2 above.

iii) Line 47: I do not think that Optical tomography can be described as a mainstream imaging modality.

Reply: Thank you for the helpful suggestion. We replaced optical tomography with optical imaging since fluorescence imaging is widely used in clinics.

iv) Penetration depth of 4 cm and resolution of around 0.4 mm is very impressive.

Reply: Thank you for the compliment.

v) Line 254: It is not clear why this statement especially about a certain value of absorption coefficient has clinical significance: The system is sufficiently sensitive to detect a 1 mm-diameter tumor with an absorption coefficient 2.1 times as high as the surrounding phantom.

Reply: Thank you for the question. Since the average absorption coefficient in human breast cancer is about 2.1 times as high as healthy breast tissue [22], we made the tumor and breast phantoms with similar optical properties as in real human breasts. The absorption coefficients μ_a of the tumor and breast phantoms were 0.105 cm⁻¹ and 0.05 cm⁻¹, respectively [22,23]. The reduced scattering coefficient μ'_s of the phantoms was 5 cm⁻¹ at 1064 nm [6, 23].

We added the above information to the manuscript on Page 14.

vi) Materials and Methods are written very elaborately. These are very useful for the interested reader in a specialist journal where it should be published.

Reply: Thank you for the compliment. We shared the technical details elaborately so that readers will learn from our experience and thus help establish standards for this technology. In addition to the photoacoustic engineers, our work has already attracted multiple preclinical researchers and clinical practitioners seeking collaborations in the past few months. In addition to the photoacoustic imaging field, we hope the work will also excite the whole optical imaging field, including scientists who need imaging technologies with functional optical contrast, deep penetration, fine spatial resolution, and high imaging speed.

References

- [1] Speckle patter. *Wikipedia* (2020). https://en.wikipedia.org/wiki/Speckle_pattern.
- [2] Hu, P., Li, L., Lin, L. & Wang, L.V. Spatiotemporal antialiasing in photoacoustic computed tomography. *IEEE Trans Med Imaging* (2020).
- [3] American national standards institute. *American National Standard for the Safe Use of Lasers ANSI, z136.131–2007* (Laser Institute of America, Orlando, FL, 2007).
- [4] Davoudi, N., Deán-Ben, X.L. & Razansky, D. Deep learning optoacoustic tomography with sparse data. *Nat Mach Intell* **1**, 453–460 (2019).
- [5] Meng, J., *et al.* High-speed, sparse-sampling three-dimensional photoacoustic computed tomography in vivo based on principal component analysis. *J Biomed Opt* **21**, 076007 (2016).
- [6] Generic tissue optical properties. https://omlc.org/news/feb15/generic_optics/index.html.
- [7] Choudhary, T.R., *et al.* Assessment of acute mild hypoxia on retinal oxygen saturation using snapshot retinal oximetry. *Investig Ophthalmol Vis Sci* **54**, 7538–7543 (2013).
- [8] Lei, L., *et al.* Label-free photoacoustic computed tomography of whole mouse brain structures ex vivo. *Neurophotonics* **3**, 035001 (2016).
- [9] Greenough, A. & Miller, A.D. *Manual of Neonatal Respiratory Care (Second Edition)*, 300–302 (2006).
- [10] Cao, R., *et al.* Functional and oxygen-metabolic photoacoustic microscopy of the awake mouse brain. *Neuroimage* **150**, 77–87 (2017).
- [11] Jasanoff, A. MRI contrast agent for functional molecular imaging of brain activity. *Curr Opin Neurobiol* **17**, 593–600 (2007).
- [12] Schwinn, D.A., McIntyre, R.W. & Reves, J.G. Isoflurane-induced vasodilation: role of the alpha-adrenergic nervous system. *Anesth Analg* **71**, 451–459 (1990).
- [13] Lyons, D.G., Parpaleix, A., Roche, M. & Charpak, S. Mapping oxygen concentration in the awake mouse brain. *Elife* **5**(2016).
- [14] Gottschalk, S., *et al.* Rapid volumetric optoacoustic imaging of neural dynamics across the mouse brain. *Nat Biomed Eng* **3**, 392-401 (2019).
- [15] Matsumoto, Y., *et al.* Visualising peripheral arterioles and venules through high-resolution and large-area photoacoustic imaging. *Sci Rep* **8**, 14930 (2018).
- [16] Oraevsky, A., *et al.* Full-view 3D imaging system for functional and anatomical screening of the breast. Proc. SPIE 10494, *Photons Plus Ultrasound: Imaging and Sensing* (2018).

- [17] Schoustra, S.M., *et al.* Twente photoacoustic mammoscope 2: system overview and three-dimensional vascular network images in healthy breasts. *J Biomed Opt* **24**, 121909 (2019).
- [18] Xu, M. & Wang, L.V. Universal back-projection algorithm for photoacoustic computed tomography. *Phys Rev E* **67**, 056605 (2003).
- [19] Pan, X., Zou, Y. & Anastasio, M.A. Data redundancy and reduced-scan reconstruction in reflectivity tomography. *IEEE Trans Image Process* **12**, 784–795 (2003).
- [20] Lin, L., *et al.* Single-breath-hold photoacoustic computed tomography of the breast. *Nat Commun* **9**, 1–9 (2018).
- [21] Contrast versus noise, eye versus machine. https://camera.hamamatsu.com/jp/en/technical_guides/contrast/index.html
- [22] Grosenick, D., Rinneberg, H., Cubeddu, R. & Taroni, P. Review of optical breast imaging and spectroscopy. *J Biomed Opt* **21**, 091311 (2016).
- [23] Durduran, T., *et al.* Bulk optical properties of healthy female breast tissue. *Phys. Med. Biol.* **47**, 2847–2861 (2002).

Reviewer #1: Authors have removed all my concerns. I have no further questions.

Reply: Thank you very much for the support.

Reviewer #2: The authors have addressed all my concerns.

Reply: Thank you very much for the support.

Reviewer #3: Authors have done a splendid job to address my comments and questions. The rebuttal and modifications have convinced me that this work should be published in the journal.

Reply: Thank you very much for the support.